# Geometry-Aware Approaches for Balancing Performance and Theoretical Guarantees in Linear Bandits

**Yuwei Luo, Mohsen Bayati**
Graduate School of Business, Stanford University
`{yuweiluo,bayati}@stanford.edu`

## Abstract

This paper is motivated by recent research in the $d$-dimensional stochastic linear bandit literature, which has revealed an unsettling discrepancy: algorithms like Thompson sampling and Greedy demonstrate promising empirical performance, yet this contrasts with their pessimistic theoretical regret bounds. The challenge arises from the fact that while these algorithms may perform poorly in certain problem instances, they generally excel in typical instances. To address this, we propose a new data-driven technique that tracks the geometric properties of the uncertainty ellipsoid around the main problem parameter. This methodology enables us to formulate a data-driven frequentist regret bound, which incorporates the geometric information, for a broad class of base algorithms, including Greedy, OFUL, and Thompson sampling. This result allows us to identify and "course-correct" problem instances in which the base algorithms perform poorly. The course-corrected algorithms achieve the minimax optimal regret of order $\widetilde{\mathcal{O}}(d\sqrt{T})$ for a $T$-period decision-making scenario, effectively maintaining the desirable attributes of the base algorithms, including their empirical efficacy. We present simulation results to validate our findings using synthetic and real data.

## 1 Introduction

Multi-armed bandits (MABs) provide a framework for studying the exploration-exploitation trade-off in sequential decision-making, where a decision-maker selects actions and observes uncertain rewards. This extends to contextual bandits with features or covariates, as shown in numerous applications (Langford & Zhang, 2008; Li et al., 2010; Tewari & Murphy, 2017; Zhou et al., 2020; Villar et al., 2015; Bastani & Bayati, 2020; Cohen et al., 2020). This paper focuses on a well-studied class of models that captures both MABs and contextual bandits as special cases while being amenable to theoretical analysis: the stochastic linear bandit (LB) problem. In this model, the problem parameter $\theta^\star$ represents an unknown vector in $\mathbb{R}^d$, while the actions, also vectors in $\mathbb{R}^d$, yield noisy rewards with a mean equal to the inner product of $\theta^\star$ and the chosen action. The objective of a policy is to maximize the cumulative reward based on the observed data up to the decision time. The policy's performance is measured by the cumulative regret, which quantifies the difference between the total expected rewards achieved by the policy and the maximum achievable expected reward.

Achieving this objective necessitates striking a balance between exploration and exploitation. In the context of LB, this entails selecting actions that aid in estimating the true parameter $\theta^\star$ accurately while obtaining optimal rewards. Various algorithms based on the *optimism principle* have been developed to address this challenge, wherein the optimal action is chosen based on the upper confidence bound (UCB) (Lai & Robbins, 1985; Auer, 2002; Dani et al., 2008; Rusmevichientong & Tsitsiklis, 2010). Another popular strategy is Thompson sampling (TS), a Bayesian heuristic introduced by Thompson (1933) that employs randomization to select actions according to the posterior distribution of reward functions. Additionally, the Greedy policy that selects the myopically best action is shown to be effective in contextual bandits (Kannan et al., 2018; Raghavan et al., 2018; Hao et al., 2020; Bastani et al., 2021).

In the linear bandit setting, two regret types are considered. Bayesian regret treats parameter $\theta^\star$ as a random variable with a prior distribution, averaging regret over noise, algorithm randomness, and parameter randomness, measuring expected performance across parameter realizations. Russo & Van Roy (2014) and Dong & Van Roy (2018) establish an $\widetilde{\mathcal{O}}(d\sqrt{T})$ upper bound for the Bayesian regret of the Thompson Sampling (TS) heuristic, referred to as LinTS, matching the minimax optimal bound by Dani et al. (2008). Here, $\widetilde{\mathcal{O}}$ denotes asymptotic order up to polylogarithmic factors. Frequentist regret assumes fixed unknown parameter $\theta^\star$, averaging only over noise and algorithm randomness. The OFUL algorithm (Abbasi-Yadkori et al., 2011) achieves an optimal $\widetilde{\mathcal{O}}(d\sqrt{T})$ frequentist regret bound. However, TS-Freq, a frequentist LinTS variant with inflated posterior variance, only achieves $\widetilde{\mathcal{O}}(d\sqrt{dT})$ (Agrawal & Goyal, 2013; Abeille et al., 2017), suboptimal by factor $\sqrt{d}$. Hamidi & Bayati (2020a) confirms this inflation is necessary and LinTS's frequentist regret cannot be improved. The Greedy algorithm lacks theoretical guarantees for linear bandit problems (Lattimore & Szepesvari, 2017), suggesting both LinTS and Greedy may perform suboptimally.

Despite the theoretical gaps, LinTS demonstrates strong empirical performance (Russo et al., 2018), suggesting posterior distribution inflation may be unnecessary in most scenarios. Similarly, the Greedy algorithm performs well in typical cases (Bietti et al., 2021). While optimism-based algorithms are computationally expensive (generally NP-hard (Dani et al., 2008; Russo & Van Roy, 2014; Agrawal, 2019)), LinTS and Greedy maintain computational efficiency. This disparity between theoretical, computational, and empirical performance prompts two questions: Can we identify problematic instances for LinTS and Greedy in a data-driven way and apply "course-correction" to ensure competitive frequentist regret bounds? Can this be achieved while preserving their empirical performance and computational efficiency? In this paper, we provide positive answers to both questions. Specifically, we make the following *contributions*.

1. We develop a real-time geometric analysis technique for the $d$-dimensional confidence ellipsoid surrounding $\theta^\star$. This method is crucial for maximizing the use of historical data, advancing beyond methods that capture only limited information from the confidence ellipsoid, such as a single numerical value. Consequently, this facilitates a more precise "course-correction".

2. We introduce a comprehensive family of algorithms, termed *POFUL* (encompassing OFUL, LinTS, TS-Freq, and Greedy as specific instances), and derive a general, data-driven frequentist regret bound for them. This bound is efficiently computable using data observed from previous decision epochs.

3. We introduce course-corrected variants of LinTS and Greedy that achieve minimax optimal frequentist regret. These adaptations maintain most of the desirable characteristics of the original algorithms.

## 1.1 OTHER RELATED LITERATURE

Our work is closely related to three main research streams: methodological foundations of linear bandits, bandit algorithms utilizing spectral properties, and data-driven exploration techniques. While these works share some similarities with our approach, we highlight the key differences and the unique aspects of our methodology.

From a methodological perspective, our regret analysis builds upon the foundations laid by Abbasi-Yadkori et al. (2011), Agrawal & Goyal (2013), and Abeille et al. (2017). However, a key distinguishing factor is that our approach does not rely on optimistic samples, which is a departure from previous methods. This means that the algorithms we study do not always choose actions that are expected to perform better than the true optimal action. By allowing non-optimistic samples, we avoid the need to inflate the posterior distribution, a requirement in the works of Agrawal & Goyal (2013) and Abeille et al. (2017).

Our use of spectral information in bandit algorithms bears some resemblance to the study of *Spectral Bandits* (Valko et al., 2014; Kocák et al., 2014; Kocák et al., 2020; Kocák & Garivier, 2020). These works represent arm rewards as smooth functions on a graph, leveraging low-rank structures to improve algorithmic performance and obtain regret guarantees independent of the number of actions. In contrast, our approach exploits the spectral properties of the action covariance matrix, which is distinct from graph spectral analysis. Moreover, our research tackles the broader context of stochastic linear bandits without assuming any low-rank structure.

Our work also shares conceptual similarities with research on exploration strategies (Russo & Van Roy, 2016; Kirschner & Krause, 2018) and data-driven exploration reduction (Bastani et al., 2021; Pacchiano et al., 2020; Hamidi & Bayati, 2020a;b). However, our methodology and data utilization differ significantly. For instance, Bastani et al. (2021) focuses on the minimum eigenvalue of the covariance matrix, a single-parameter summary of the observed data, while Hamidi & Bayati (2020b) uses information from one-dimensional reward confidence intervals. The work of Hamidi & Bayati (2020a) is more closely related to ours, as it employs spectral information to improve the performance of Thompson Sampling in linear bandits. They use a single summary statistic called the *thinness coefficient* to decide whether to inflate the posterior. In contrast, our approach leverages the full geometric details of the $d$-dimensional confidence ellipsoid, harnessing richer geometric information.

## 2 SETUP AND PRELIMINARIES

**Notations.** We use $\|\cdot\|$ to denote the Euclidean 2-norm. For a symmetric positive definite matrix $A$ and a vector $x$ of proper dimension, we let $\|x\|_A := \sqrt{x^\top A x}$ be the weighted 2-norm (or the $A$-norm). We let $\langle\cdot,\cdot\rangle$ denote the inner product in Euclidean space such that $\langle x, y\rangle = x^\top y$. For a $d$-dimensional matrix $V$, we let $\lambda_1(V) \geq \lambda_2(V) \geq \cdots \geq \lambda_d(V)$ be the eigenvalues of $V$ arranged in decreasing order. We let $\mathcal{B}_d$ denote the unit ball in $\mathbb{R}^d$, and $\mathcal{S}_{d-1} = \{x \in \mathbb{R}^d : \|x\| = 1\}$ denote the unit hypersphere in $\mathbb{R}^d$. For an interger $N \geq 1$, we let $[N]$ denote the set $\{1, 2, \ldots, N\}$. We use the $\mathcal{O}(\cdot)$ notation to suppress problem-dependent constants, and the $\widetilde{\mathcal{O}}(\cdot)$ notation further suppresses polylog factors.

**Problem formulation and assumptions.** We consider the stochastic linear bandit problem. Let $\theta^\star \in \mathbb{R}^d$ be a fixed but unknown parameter. At each time $t \in [T]$, a policy $\pi$ selects action $x_t$ from a set of action $\mathcal{X}_t \subset \mathbb{R}^d$ according to the past observations and receives a reward $r_t = \langle x_t, \theta^\star\rangle + \varepsilon_t$, where $\varepsilon_t$ is mean-zero noise with a distribution specified in Assumption 3 below. We measure the performance of $\pi$ with the cumulative expected regret $\mathcal{R}(T) = \sum_{t=1}^{T}\langle x_t^\star, \theta^\star\rangle - \langle x_t, \theta^\star\rangle$, where $x_t^\star$ is the best action at time $t$, i.e., $x_t^\star = \arg\max_{x \in \mathcal{X}_t}\langle x, \theta^\star\rangle$. Let $\mathcal{F}_t$ be a $\sigma$-algebra generated by the history $(x_1, r_1, \ldots, x_t, r_t)$ and the prior knowledge, $\mathcal{F}_0$. Therefore, $\{\mathcal{F}_t\}_{t\geq 0}$ forms a filteration such that each $\mathcal{F}_t$ encodes all the information up to the end of period $t$.

We make the following assumptions that are standard in the relevant literature.

**Assumption 1** (Bounded parameter). *The unknown parameter $\theta^\star$ is bounded as $\|\theta^\star\| \leq S$, where $S > 0$ is known.*

**Assumption 2** (Bounded action sets). *The action sets $\{\mathcal{X}_t\}$ are uniformly bounded and closed subsets of $\mathbb{R}^d$, such that $\|x\| \leq X_t$ for all $x \in \mathcal{X}_t$ and all $t \in [T]$, where $X_t$'s are known and $\sup_{t\geq 1}\{X_t\} < \infty$.*

**Assumption 3** (Subgaussian reward noise). *The noise sequence $\{\varepsilon_t\}_{t\geq 1}$ is conditionally mean-zero and $R$-subgaussian, where $R$ is known. Formally, for all real valued $\lambda$, $\mathbb{E}\left[e^{\lambda\varepsilon_t}|\mathcal{F}_t\right] \leq \exp\left(\lambda^2 R^2/2\right)$. This condition implies that $\mathbb{E}\left[\varepsilon_t|\mathcal{F}_t\right] = 0$ for all $t \geq 1$.*

### 2.1 REGULARIZED LEAST SQUARE AND CONFIDENCE ELLIPSOID

In this subsection, we review the useful frequentist tools developed by Abbasi-Yadkori et al. (2011) for estimating the unknown parameter $\theta^*$ in linear bandit (LB) problems.

Consider an arbitrary sequence of actions $(x_1, \ldots, x_t)$ and their corresponding rewards $(r_1, \ldots, r_t)$. In LB problems, the parameter $\theta^*$ is typically estimated using the regularized least squares (RLS) estimator. Let $\lambda_{\mathrm{reg}}$ be a fixed regularization parameter. The sample covariance matrix $V_t$ and the RLS estimate $\widehat{\theta}_t$ are defined as follows:

$$V_t = \lambda_{\mathrm{reg}}I_d + \sum_{s=1}^{t}x_s x_s^\top, \quad \widehat{\theta}_t = V_t^{-1}\sum_{s=1}^{t}x_s r_s. \tag{1}$$

The following proposition from Abbasi-Yadkori et al. (2011) establishes that the RLS estimate $\widehat{\theta}_t$ concentrates around the true parameter $\theta^*$ with high probability.

**Proposition 1** (Theorem 2 in Abbasi-Yadkori et al. (2011)). *Let $\delta \in (0, 1)$ be a fixed confidence level. Then, with probability at least $1 - \delta$, it holds for all $x \in \mathbb{R}^d$ that*

$$\|\widehat{\theta}_t - \theta^\star\|_{V_t} \leq \beta_{t,\delta,\lambda_{reg}}^{RLS}, \quad |\langle x, \widehat{\theta}_t - \theta^\star\rangle| \leq \|x\|_{V_t^{-1}}\beta_{t,\delta,\lambda_{reg}}^{RLS}$$

*where the confidence bound $\beta_{t,\delta,\lambda_{reg}}^{RLS}$ is defined as*

$$\beta_{t,\delta,\lambda_{reg}}^{RLS} = R\sqrt{2\log(\lambda_{reg} + t)^{d/2}\lambda_{reg}^{-d/2}\delta^{-1}} + \sqrt{\lambda_{reg}}S. \tag{2}$$

Proposition 1 enables us to construct the following sequence of confidence ellipsoids.

**Definition 1.** *Fix $\delta \in (0, 1)$. We define the RLS confidence ellipsoid as*

$$\mathcal{E}_{t,\delta,\lambda_{reg}}^{RLS} = \{\theta \in \mathbb{R}^d : \|\theta - \widehat{\theta}_t\|_{V_t} \leq \beta_{t,\delta,\lambda_{reg}}^{RLS}\}.$$

The next proposition, known as the *elliptical potential lemma*, plays a central role in bounding the regret. This proposition provides the key element in the work of Abbasi-Yadkori et al. (2011), showing that the cumulative prediction error incurred by the action sequence used to estimate $\theta^*$ is small.

**Proposition 2** (Lemma 11 in Abbasi-Yadkori et al. (2011)). *If $\lambda_{reg} > 1$, for an arbitrary sequence $(x_1, \ldots, x_t)$, it holds that $\sum_{s=1}^{t}\|x_s\|_{V_s^{-1}}^2 \leq 2\log\frac{\det(V_{t+1})}{\det(\lambda_{reg}I)} \leq 2d\log(1 + \frac{t}{\lambda_{reg}})$.*

## 3 POFUL ALGORITHMS

In this section, we introduce POFUL (Pivot OFUL), a generalized framework of OFUL. This framework enables a unified analysis of frequentist regret for common algorithms.

At a high level, POFUL is designed to encompass the exploration mechanism of OFUL and LinTS. POFUL takes as input a sequence of *inflation* parameters $\{\iota_t\}_{t\in[T]}$, feasible (randomized) *pivots* $\{\widetilde{\theta}_t\}_{t\in[T]}$ and *optimism* parameters $\{\tau_t\}_{t\in[T]}$. The inflation parameters are used to construct confidence ellipsoids that contain $\{\widetilde{\theta}_t\}_{t\in[T]}$ with high probability. This is formalized in the next definition.

**Definition 2.** *Fix $\delta \in (0, 1)$ and $\delta' = \delta/2T$. Given the inflation parameters $\{\iota_t\}_{t\in[T]}$, we call random variables $\{\widetilde{\theta}_t\}_{t\in[T]}$ feasible pivots if for all $t \in [T]$, $\mathbb{P}[\widetilde{\theta}_t \in \mathcal{E}_{t,\delta',\lambda_{reg}}^{PVT}|\mathcal{F}_t] \geq 1 - \delta'$, where we define the "pivot ellipsoid" as $\mathcal{E}_{t,\delta,\lambda_{reg}}^{PVT} := \{\theta \in \mathbb{R}^d : \|\theta - \widehat{\theta}_t\|_{V_t} \leq \iota_t\beta_{t,\delta,\lambda_{reg}}^{RLS}\}$.*

At each time $t$, POFUL chooses the action that maximizes the optimistic reward

$$\widetilde{x}_t = \arg\max_{x\in\mathcal{X}_t} \langle x, \widetilde{\theta}_t\rangle + \tau_t\|x\|_{V_t^{-1}}\beta_{t,\delta',\lambda_{reg}}^{RLS}, \tag{3}$$

as shown in a pseudocode representation in Algorithm 1 and illustrated in Figure 1a.

Recall OFUL encourages exploration by introducing the uncertainty term $\tau_t\|x\|_{V_t^{-1}}\beta_{t,\delta',\lambda_{reg}}^{RLS}$ in the reward, while LinTS explores through random sampling within the confidence ellipsoid. We let POFUL select an arbitrary pivot (which can be random) from $\mathcal{E}_{t,\delta',\lambda_{reg}}^{PVT}$ and maximize the optimistic reward to encompass arbitrary exploration mechanisms within $\mathcal{E}_{t,\delta',\lambda_{reg}}^{PVT}$.

We demonstrate that POFUL encompasses OFUL, LinTS, TS-Freq, and Greedy as special cases, as illustrated in Figure 1b.

**Example 1** (OFUL). *For stochastic linear bandit problems, OFUL chooses actions by solving the optimization problem $\max_{x\in\mathcal{X}_t} \langle x, \widehat{\theta}_t\rangle + \|x\|_{V_t^{-1}}\beta_{t,\delta',\lambda_{reg}}^{RLS}$. Therefore, OFUL is a specially case of POFUL where $\iota_t = 0$, $\tau_t = 1$ and $\widetilde{\theta}_t = \widehat{\theta}_t$, the center of the confidence ellipsoid, for all $t \in [T]$.*

Before describing how TS can be derived as an instance of POFUL, we introduce a definition.

**Definition 3.** *Let $\delta \in (0, 1)$. We define $\mathcal{D}^{SA}(\delta)$ as a distribution satisfying $\mathbb{P}_{\eta\sim\mathcal{D}^{SA}(\delta)}[\|\eta\| \leq 1] \geq 1 - \delta$.*

---

**Algorithm 1** POFUL

---

**Require:** $T, \delta, \lambda_{\text{reg}}, \{\iota_t\}_{t\in[T]}, \{\tau_t\}_{t\in[T]}$

  Initialize $V_0 \leftarrow \lambda_{\text{reg}} I_d, \widehat{\theta}_1 \leftarrow 0, \delta' \leftarrow \delta/2T$

  **for** t = 0, 1, ..., T **do**

    Sample a feasible pivot $\widetilde{\theta}_t$ with respect to $\iota_t$ according to Definition 2

    $\widetilde{x}_t \leftarrow \arg\max_{x\in\mathcal{X}_t}\langle x, \widetilde{\theta}_t\rangle + \tau_t\|x\|_{V_t^{-1}}\beta^{RLS}_{t,\delta',\lambda_{\text{reg}}}$

    Observe reward $r_t$

    $V_{t+1} \leftarrow V_t + \widetilde{x}_t\widetilde{x}_t^\top$

    $\widehat{\theta}_{t+1} \leftarrow V_{t+1}^{-1}\sum_{s=1}^t \widetilde{x}_s r_s.$

  **end for**

---

**Example 2** (TS). *Linear Thompson Sampling (LinTS) algorithm is a generic randomized algorithm that samples from a distribution constructed from the RLS estimate at each step. At time $t$, LinTS samples as $\widetilde{\theta}_t = \widehat{\theta}_t + \iota_t^{TS}\beta^{RLS}_{t,\delta',\lambda_{reg}}V_t^{-1/2}\eta_t$, where $\delta' = \delta/2T$, $\iota_t^{TS}$ is inflation parameter controlling the scale of the sampling range, and $\eta_t$ is a random sample from a normalized sampling distribution $\mathcal{D}^{SA}(\delta')$ that concentrates with high probability. LinTS is a special case of POFUL where $\iota_t = \iota_t^{TS}$, $\tau_t = 0$ and $\widetilde{\theta}_t = \widehat{\theta}_t + \iota_t^{TS}\beta^{RLS}_{t,\delta',\lambda_{reg}}V_t^{-1/2}\eta_t$. Setting $\iota_t = \widetilde{\mathcal{O}}(1)$ corresponds to the original LinTS algorithm, while setting $\iota_t = \widetilde{\mathcal{O}}(\sqrt{d})$ corresponds to the frequentist variant of LinTS studied in Agrawal & Goyal (2013); Abeille et al. (2017), namely TS-Freq. This means TS-Freq inflates the posterior by a factor of order $\sqrt{d}$.*

**Example 3** (Greedy). *Greedy is a special case of POFUL with $\iota_t = \tau_t = 0$, $\widetilde{\theta}_t = \widehat{\theta}_t$, $\forall t$.*

## 4 FREQUENTIST REGRET ANALYSIS OF POFUL

In this section, we present the frequentist regret analysis of POFUL algorithms. Details of the proofs are deferred to the longer version of the paper Luo & Bayati (2023). We first introduce high-probability concentration events.

**Definition 4.** *Fix $\delta \in (0,1)$ and $\delta' = \delta/2T$. We define $\beta^{PVT}_{t,\delta',\lambda_{reg}} := \iota_t\beta^{RLS}_{t,\delta',\lambda_{reg}}$, $\widehat{\mathcal{A}}_t := \{\forall s \leq t : \|\widehat{\theta}_t - \theta^\star\|_{V_t} \leq \beta^{RLS}_{t,\delta',\lambda_{reg}}\}$, $\widetilde{\mathcal{A}}_t := \{\forall s \leq t : \|\widetilde{\theta}_t - \widehat{\theta}_t\|_{V_t} \leq \beta^{PVT}_{t,\delta',\lambda_{reg}}\}$, and $\mathcal{A}_t := \widehat{\mathcal{A}}_t \cap \widetilde{\mathcal{A}}_t$.*

**Proposition 3.** *Under Assumptions 1, 2 and 3 , we have $\mathbb{P}[\mathcal{A}_T] \geq 1 - \delta$.*

### 4.1 AN DATA-DRIVEN REGRET BOUND FOR POFUL

In the following, we condition on the event $\mathcal{A}_T$ which holds with probability $1 - \delta$. The following proposition bounds the instantaneous regret of POFUL.

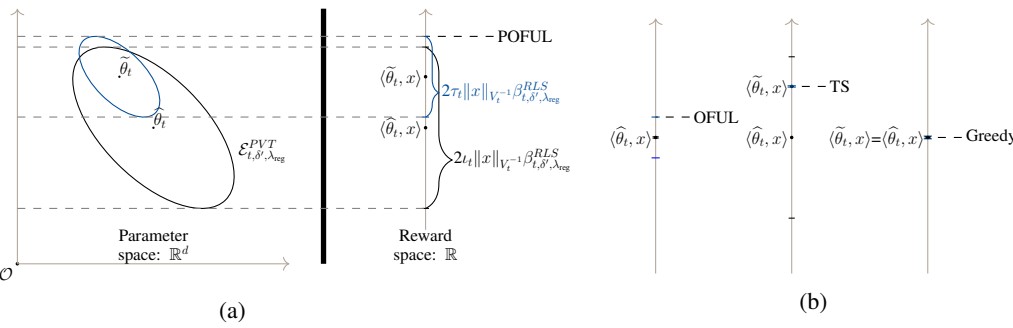

Figure 1: (a) POFUL algorithms illustration for general $\iota_t$ and $\tau_t$. (b) Special cases: OFUL ($\iota_t = 0$, $\tau_t = 1$), TS ($\tau_t = 0$), and Greedy ($\iota_t = \tau_t = 0$).

**Proposition 4.** *Suppose $\theta^\star \in \mathcal{E}^{RLS}_{t,\delta',\lambda_{reg}}$ and $\widetilde{\theta}_t \in \mathcal{E}^{PVT}_{t,\delta',\lambda_{reg}}$, it holds that*

$$\langle x_t^\star, \theta^\star \rangle - \langle \widetilde{x}_t, \theta^\star \rangle \le (1 + \iota_t - \tau_t)\|x_t^\star\|_{V_t^{-1}}\beta^{RLS}_{t,\delta',\lambda_{reg}} + (1 + \iota_t + \tau_t)\|\widetilde{x}_t\|_{V_t^{-1}}\beta^{RLS}_{t,\delta',\lambda_{reg}}. \qquad (4)$$

Note that this upper bound is different from what's used in the optimism-based methods (Abbasi-Yadkori et al., 2011; Agrawal & Goyal, 2013; Abeille et al., 2017), we reproduce their upper bound and discuss the relationship of our method and theirs in the longer version of the paper Luo & Bayati (2023).

On the right-hand side of equation 4, since the oracle optimal action sequence $\{x_t^\star\}_{t \in [T]}$ is unknown to the algorithm and is different from the action sequence $\{\widetilde{x}_t\}_{t \in [T]}$ played by POFUL, one cannot apply Proposition 2 to bound the summation $\sum_{t=1}^{T}\|\widetilde{x}_t\|^2_{V_t^{-1}}$ and get an upperbound of the regret. To address this, the key point to connect $\{\widetilde{x}_t\}_{t \in [T]}$ and $\{x_t^\star\}_{t \in [T]}$ with $V_t^{-1}$-norm. This motivates the following definition.

**Definition 5.** *For each $t \ge 1$, let $\widetilde{x}_t$ and $x_t^\star$ respectively denote the action chosen by POFUL and the optimal action. We define the uncertainty ratio at time $t$ as $\alpha_t := \|x_t^\star\|_{V_t^{-1}}/\|\widetilde{x}_t\|_{V_t^{-1}}$. We also define the (instantaneous) regret proxy at time $t$ as $\mu_t := \alpha_t(1 + \iota_t - \tau_t) + 1 + \iota_t + \tau_t$.*

Note that $\langle x, \widehat{\theta}_t - \theta^\star \rangle \le \|x\|_{V_t^{-1}}\beta^{RLS}_{t,\delta',\lambda_{reg}}$ holds with high probability, we have that $\|x\|_{V_t^{-1}}$ essentially determines the length of the confidence interval of the reward $\langle x, \theta^\star \rangle$. Hence, $\alpha_t$ serves as the ratio of uncertainty degrees of the reward obtained by the optimal action $x_t^\star$ and the chosen action $\widetilde{x}_t$.

The intuition behind the definition for $\mu_t$ is constructing a regret upper bound similar to that of OFUL. Specifically, Proposition 4 indicates $\langle x_t^\star, \theta^\star \rangle - \langle \widetilde{x}_t, \theta^\star \rangle \le \mu_t\|\widetilde{x}_t\|_{V_t^{-1}}\beta^{RLS}_{t,\delta',\lambda_{reg}}$, and we can check that the instantaneous regret of OFUL satisfies $\langle x_t^\star, \theta^\star \rangle - \langle \widetilde{x}_t, \theta^\star \rangle \le 2\|\widetilde{x}_t\|_{V_t^{-1}}\beta^{RLS}_{t,\delta',\lambda_{reg}}$. In this sense, $\mu_t$ is a proxy of the instantaneous regret incurred by POFUL at time $t$. Moreover, OFUL can be regarded as a POFUL algorithm whose $\mu_t$ is fixed at 2, and we could extend the definition of $\alpha_t$ to OFUL by solving $\mu_t = \alpha_t(1 + \iota_t - \tau_t) + 1 + \iota_t + \tau_t$ and set $\alpha_t = 1$ for all $t \in [T]$ for OFUL (recall that in OFUL, $\iota_t = 0$ and $\tau_t = 1$ for all $t \in [T]$).

The following Theorem connects $\{\mu_t\}_{t \in [T]}$ and $\mathcal{R}(T)$. It provides an oracle but general frequentist regret upper bound for all POFUL algorithms.

**Theorem 1** (Oracle frequentist regret bound for POFUL)**.** *Fix $\delta \in (0,1)$ and let $\delta' = \delta/2T$. Under Assumptions 1, 2 and 3, with probability $1 - \delta$, POFUL achieves a regret of*

$$\mathcal{R}(T) \le \sqrt{2d\left(\sum_{t=1}^{T}\mu_t^2\right)\log\left(1 + \frac{T}{\lambda_{reg}}\right)}\beta^{RLS}_{T,\delta',\lambda_{reg}}. \qquad (5)$$

**Remark 1.** *We call Theorem 1 an oracle regret bound as $\{\mu_t\}_{t \in [T]}$ for general POFUL depends on the unknown system parameter $\theta^\star$. In general, they cannot be calculated by the decision-maker. Nevertheless, note that $\iota_t$ and $\tau_t$ are chosen by the decision-maker, when we have computable upper bounds $\{\widehat{\alpha}_t\}_{t \in [T]}$ for $\{\alpha_t\}_{t \in [T]}$, using $\mu_t^2 \le 2\alpha_t^2(1 + \iota_t - \tau_t)^2 + 2(1 + \iota_t + \tau_t)^2$, we could calculate upper bounds for $\{\mu_t\}_{t \in [T]}$ as well. Consequently, Theorem 1 instantly turns into a data-driven regret bound for POFUL and could be utilized later for course correction, which will be the aim of the next section. When we additionally know that $1 + \iota_t - \tau_t$ is non-negative, we would use the equality $\mu_t = \alpha_t(1 + \iota_t - \tau_t) + 1 + \iota_t + \tau_t$ directly for the bound.*

**Remark 2.** *In the Discussion section of Abeille et al. (2017), the authors introduce a concept similar to the reciprocal of our $\alpha_t$. They suggest that the necessity of proving LinTS samples are optimistic could be bypassed if for some $\alpha > 0$ LinTS samples $\widetilde{\theta}_t$ such that $\|x^\star(\widetilde{\theta}_t)\|_{V_t^{-1}} \ge \alpha\|x^\star(\theta_t^\star)\|_{V_t^{-1}}$ with constant probability, where $x^\star(\widetilde{\theta}_t)$ and $x^\star(\theta_t^\star)$ represent the optimal actions corresponding to $\widetilde{\theta}_t$ and $\theta_t^\star$, respectively. They pose this as an open question regarding the possibility of relaxing the requirement of inflating the posterior. In the following section, we provide a positive answer to this question by studying the reciprocal of their $\alpha$ using geometric arguments. This investigation offers an explanation for the empirical success of LinTS without the need for posterior inflation.*

## 5 A DATA-DRIVEN APPROACH

In this section, we present the main contribution of this work which provides a data-driven approach to calibrating POFUL. Note that $\iota_t$ and $\tau_t$ are parameters of POFUL that can be controlled by a decision-maker, the essential point is to find a computable, non-trivial upper bound $\widehat{\alpha}_t$ for the uncertainty ratio $\alpha_t$, which turns into an upper bound $\widehat{\mu}_t$ for the regret proxy $\mu_t$ that's deeply related to the frequentist regret of POFUL.

We focus on scenarios where $\tau_t = 0$ for all $t \in [T]$. These include LinTS and variants such as TS-Freq, as well as Greedy - standard algorithms still lacking theoretical regret guarantees. Below, we construct upper bounds $\{\widehat{\alpha}_t\}_{t \in [T]}$ for the continuous-action scenario. Bounds for discrete-action scenarios appear in the longer version of the paper Luo & Bayati (2023).

### 5.1 CONTINUOUS ACTION SPACE

Our strategy capitalizes on geometric insights related to the properties of the confidence ellipsoids, providing upper bounds that can be computed efficiently. For the sake of a better illustration, we consider $\mathcal{X}_t = \mathcal{S}_{d-1}$ for all $t \in [T]$ for this scenario, where $\mathcal{S}_{d-1} = \{x \in \mathbb{R}^d : \|x\| = 1\}$ is the unit hypersphere in $\mathbb{R}^d$. This is a standard example of continuous action space, and is the same as the setting considered in Abeille et al. (2017). We remark that for this specific setting, the problem is still hard. This is because we don't have a closed-form solution for the set of potentially optimal actions.

In this setting, the optimal action $x_t^\star(\theta) := \arg\max_{x \in \mathcal{X}_t} \langle x, \theta \rangle$ takes the form $x_t^\star(\theta) = \theta/\|\theta\|$. To upper bound $\alpha_t$, we consider respectively the smallest and largest value of $\|x_t^\star(\theta)\|_{V_t^{-1}}$ for $\theta$ in the confidence ellipsoids of $\theta$, namely, $\mathcal{E}_{t,\delta',\lambda_{\text{reg}}}^{RLS}$ and $\mathcal{E}_{t,\delta',\lambda_{\text{reg}}}^{PVT}$. Specifically, we have

$$\alpha_t \leq \frac{\sup_{\theta \in \mathcal{E}_{t,\delta',\lambda_{\text{reg}}}^{RLS}} \|x_t^\star(\theta)\|_{V_t^{-1}}}{\inf_{\theta \in \mathcal{E}_{t,\delta',\lambda_{\text{reg}}}^{PVT}} \|x_t^\star(\theta)\|_{V_t^{-1}}}. \tag{6}$$

As is illustrated in Figure 2, the set of potentially optimal actions $\mathcal{C}_t$ is the projection of the confidence ellipsoid $\mathcal{E}_t$ onto $\mathcal{S}_{d-1}$. It's hard to get a closed-form expression for $\mathcal{C}_t$, so we cannot directly calculate the range of $V_t^{-1}$-norm of actions in $\mathcal{C}_t$. Nevertheless, when POFUL has implemented sufficient exploration so that $\mathcal{E}_t$ is small enough, $\mathcal{C}_t$ concentrates accordingly to a small cap on $\mathcal{S}_{d-1}$. Therefore, it is possible to estimate the range of the $V_t$-norm by employing geometric reasoning. Subsequently, this estimated range will be utilized to ascertain the range of the $V_t^{-1}$-norm.

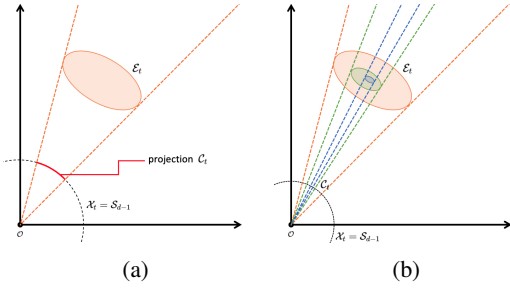

|     |     |
| :-: | :-: |
| (a) | (b) |

Figure 2: Illustration of potentially optimal actions set $\mathcal{C}_t$ in $\mathbb{R}^2$. (a): $\mathcal{C}_t$ is $\mathcal{E}_t$'s projection onto $\mathcal{S}_{d-1}$. (b): As more data is collected, $\mathcal{E}_t$ shrinks (colors show exploration levels). Potentially optimal actions point in similar directions, determining their $V_t$-norm. This suggests their $V_t$-norm range could be estimated geometrically.

The main theorem derives an upper bound for $\alpha_t$ based on this idea.

**Theorem 2.** *Suppose* $\mathcal{X}_t = \mathcal{S}_{d-1}$ *for all* $t \in [T]$. *Define* $m_t = (\|\widehat{\theta}_t\|_{V_t}^2 - (\beta_{t,\delta',\lambda_{\text{reg}}}^{RLS})^2)/(\|\widehat{\theta}_t\| + \beta_{t,\delta',\lambda_{\text{reg}}}^{RLS}/\lambda_d(V_t))^2$, $M_t = \|\widehat{\theta}_t\|_{V_t^2}^2/(\|\widehat{\theta}_t\|_{V_t}^2 - (\beta_{t,\delta',\lambda_{\text{reg}}}^{PVT})^2)$. *Let* $k \in [d]$ *be the integer that satisfies*

$\lambda_k(V) \leq M_t \leq \lambda_{k+1}(V)$. *Define* $\beta_{t,\delta',\lambda_{reg}}^{PVT} := \iota_t \beta_{t,\delta',\lambda_{reg}}^{RLS}$ *and*

$$
\Phi_t = \begin{cases} (\lambda_1^{-1}(V_t) + \lambda_d^{-1}(V_t) - m_t\lambda_1^{-1}(V_t)\lambda_d^{-1}(V_t))^{\frac{1}{2}}, & \text{if } \|\widehat{\theta}_t\|_{V_t} \geq \beta_{t,\delta',\lambda_{reg}}^{RLS}, \\ \lambda_d^{-\frac{1}{2}}(V_t), & \text{if } \|\widehat{\theta}_t\|_{V_t} < \beta_{t,\delta',\lambda_{reg}}^{RLS}, \end{cases},
$$

$$
\Psi_t = \begin{cases} (\lambda_k^{-1}(V_t) + \lambda_{k+1}^{-1}(V_t) - M_t\lambda_k^{-1}(V_t)\lambda_{k+1}^{-1}(V_t))^{\frac{1}{2}}, & \text{if } \|\widehat{\theta}_t\|_{V_t} \geq \beta_{t,\delta',\lambda_{reg}}^{PVT}, \\ \lambda_1^{-\frac{1}{2}}(V_t), & \text{if } \|\widehat{\theta}_t\|_{V_t} < \beta_{t,\delta',\lambda_{reg}}^{PVT}. \end{cases}
$$

*Then for all $t \in [T]$, conditioned on $\widehat{\mathcal{A}}_t \cap \widetilde{\mathcal{A}}_t$, it holds for all $s \leq t$ that $\alpha_s \leq \widehat{\alpha}_s := \Phi_s/\Psi_s$.*

To better understand what Theorem 2 implies, we discuss some special cases and provide empirical validations for them in the longer version of the paper Luo & Bayati (2023).

## 6  A META-ALGORITHM FOR COURSE-CORRECTION

This section demonstrates how the data-driven regret bound can enhance standard bandit algorithms. We propose a meta-algorithm that creates course-corrected variants of base algorithms, achieving minimax-optimal frequentist regret guarantees while preserving most original characteristics, including computational efficiency and typically low regret.

We take LinTS as an example of the base algorithm, and propose the algorithm Linear Thompson Sampling with Maximum Regret (Proxy) (TS-MR). The idea is to measure the performance of LinTS using $\widehat{\mu}_t$ and avoid bad LinTS actions by switching to OFUL actions. Specifically, at each time $t$, TS-MR calculates the upper bound $\widehat{\mu}_t$ and compares it with a preset threshold $\mu$. If $\widehat{\mu}_t > \mu$, LinTS might be problematic and TS-MR takes an OFUL action to ensure a low instantaneous regret; if $\widehat{\mu}_t \leq \mu$, TS-MR takes the LinTS action. We remark that setting $\iota_t = 0$ for all $t \in [T]$ yields the corresponding Greedy-MR algorithm. The pseudocode is presented in the longer version of the paper Luo & Bayati (2023).

**Remark 3.** *Computing $\widehat{\mu}_t$ primarily requires SVD decomposition of the sample covariance matrix $V_t$. Since $V_t = \lambda_{reg}I_d + \sum_{s=1}^{t} x_s x_s^\top$ is updated via rank-one matrices, its SVD can be efficiently updated (Gandhi & Rajgor, 2017), preventing computational bottlenecks.*

By design, course-corrected algorithms ensure that $\mu_t \leq \max\{\mu, 2\}$ for all $t \in [T]$. Substituting this upper bound into Theorem 1 establishes that these algorithms achieve optimal frequentist regret, up to a constant factor.

**Corollary 1.** *TS-MR and Greedy-MR achieve a frequentist regret of $\widetilde{\mathcal{O}}(\max\{\mu, 2\}d\sqrt{T})$.*

In high-risk settings where LinTS and Greedy may fail, a small $\mu$ ensures TS-MR and Greedy-MR select more OFUL actions, promoting sufficient exploration. Conversely, in low-risk settings where the original algorithms perform well, a large $\mu$ favors TS and greedy actions, minimizing unnecessary exploration and reducing computational cost. In the longer version of the paper Luo & Bayati (2023), we show how $\mu$ impacts the fraction of OFUL actions in TS-MR and Greedy-MR and their performance. Results indicate that course-corrected algorithms maintain robustness across a range of moderate $\mu$ values, suggesting that the precise selection of $\mu$ is unlikely to present a significant practical concern.

## 7  SIMULATIONS

We aim to compare TS-MR, Greedy-MR, and key baseline algorithms, via simulation.

### 7.1  SYNTHETIC DATASETS

We conduct simulations on three representative synthetic examples. We average simulation results over 100 independent runs for each of the examples. The results are shown in Figure 3.

**Example 1. Stochastic linear bandit with uniformly and independently distributed actions.** We fix $d = 50$, and sample $\theta^\star \sim \text{Unif}(\{\theta \in \mathbb{R}^d | \|\theta\| = 10\})$ on a sphere with fixed norm. At each time $t$, we generate 100 i.i.d. random actions sampled from $\text{Unif}(\mathcal{S}_{d-1})$ to form $\mathcal{X}_t$. This is a basic example of standard stochastic linear bandit problems without any extra structure. We set the threshold $\mu = 8$ for TS-MR and Greedy-MR. TS-Freq shows pessimistic regret due to the inflation of the posterior, while other algorithms in general perform well.

**Example 2. Contextual bandits embedded in the linear bandit problem (Abbasi-Yadkori, 2013).** We fix $d = 50$, and sample $\theta^\star \sim \text{Unif}(\{\theta \in \mathbb{R}^d | \|\theta\| = 70\})$. At each time $t$, we first generate a random vector $u_t \sim \mathcal{N}(0, I_5)$ and let $x_{t,i} \in \mathbb{R}^{50}$ be the vector whose $i$-th block of size 5 is a copy of $u_t$, and other components are 0. Then $X_t = \{x_{t,i}\}_{i \in [10]}$ is an action set of size 10, sharing the same feature $u_t$ in different blocks. This problem is equivalent to a 10-armed contextual bandit. We set $\mu = 12$ for TS-MR and Greedy-MR. In this setting, Greedy performs suboptimally due to a lack of exploration for some arms. Nevertheless, Greedy-MR outperforms both Greedy and OFUL by adaptively choosing OFUL actions only when it detects large regret proxy $\widehat{\mu}_t$.

**Example 3. Prior mean mismatch (Hamidi & Bayati, 2020a).** This is an example in which LinTS is shown to incur linear Bayesian regret. We sample $\theta^\star \sim \mathcal{N}(m\mathbf{1}_{3d}, I_{3d})$ and fix the action set $\mathcal{X}_t = \{0, x_a, x_b\}$ for all $t \in [T]$, where $x_a = -\sum_{i=1}^d e_i/\sqrt{3d}$, $x_b = \sum_{i=11}^{3d} e_i \sqrt{3d} - \sum_{i=1}^d e_i/\sqrt{3d}$. It is shown in Hamidi & Bayati (2020a) that, when LinTS takes a wrong prior mean as input, it has a large probability to choose $\widetilde{x}_2 = 0$, conditioned on $\widetilde{x}_1 = x_a$. Note that choosing the zero action brings no information update to LinTS, it suffers a linear Bayesian regret when trying to escape from the zero action. We let $m = 10$ and set $d = 10$, so the problem is a 30-dimensional linear bandit. We set $\mu = 12$ for TS-MR and Greedy-MR. We see both LinTS and Greedy incur linear regrets as expected, while TS-MR and Greedy-MR, switch to OFUL adaptively to tackle this hard problem and achieve sublinear regret.

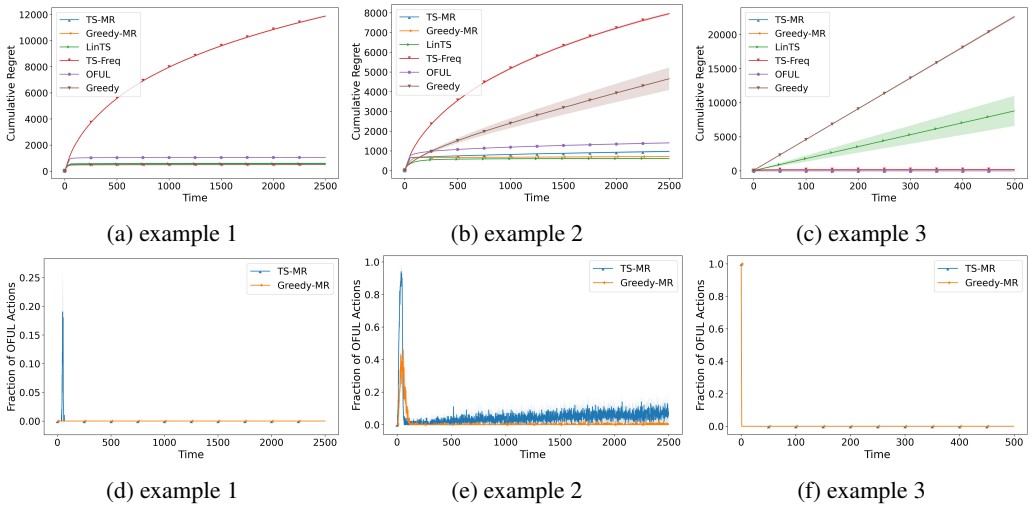

(a) example 1      (b) example 2      (c) example 3

(d) example 1      (e) example 2      (f) example 3

Figure 3: Simulation results on synthetic data. (a) - (c): Cumulative regret of TS-MR and Greedy-MR versus baseline algorithms. Shaded regions show $\pm 2$ SE of mean regret. (d) - (f): Fraction of OFUL actions in TS-MR and Greedy-MR.

## 7.2 REAL-WORLD DATASETS

We explore the performance of standard POFUL algorithms and the proposed TS-MR and Greedy-MR algorithms on real-world datasets. We use three classification datasets from OPENML: Cardiotocography, JapaneseVowels, and Segment, representing healthcare, pattern recognition, and computer vision domains. Following Bietti et al. (2021); Bastani et al. (2021), we convert these classification tasks to contextual bandit problems and embed them into linear bandit problems as in

Example 2, Section 7.1. Each class becomes an action where the decision-maker receives a binary reward (1 for correct classification, 0 otherwise) plus Gaussian noise.

We plot the cumulative regret (averaged over 100 runs) for all algorithms. Figure 4 shows that for all real-world datasets: OFUL and TS-Freq perform poorly due to their conservative exploration; LinTS and Greedy are achieving empirical success even though they don't have theoretical guarantees; TS-MR and Greedy-MR retain the desirable empirical performance of LinTS and Greedy, while enjoying the minimax optimal frequentist regret bound.

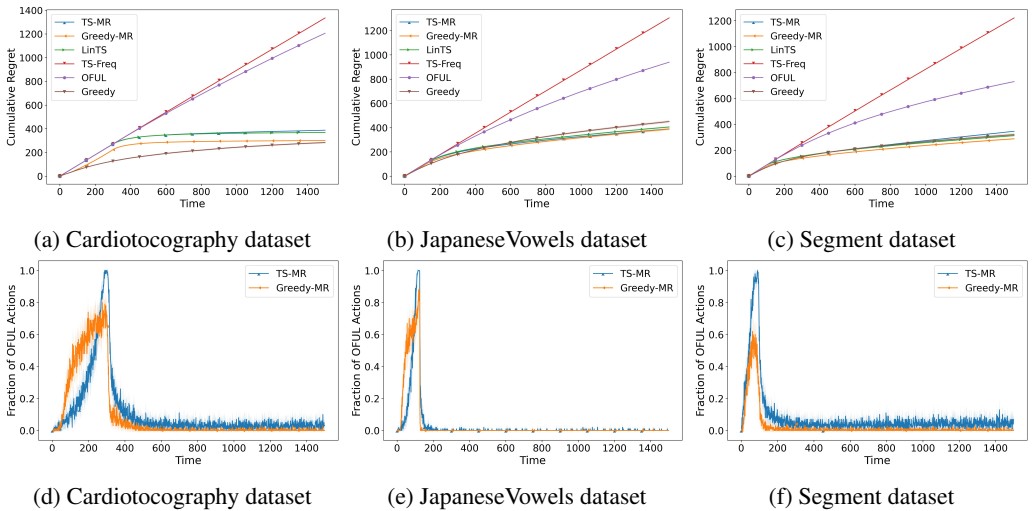

(a) Cardiotocography dataset    (b) JapaneseVowels dataset    (c) Segment dataset

(d) Cardiotocography dataset    (e) JapaneseVowels dataset    (f) Segment dataset

Figure 4: Simulation results on real-world datasets. (a) - (c): Cumulative regret of all algorithms. Shaded regions show the $\pm 2$ SE of the mean regret. (d) - (f): Fraction of OFUL actions of TS-MR and Greedy-MR.

**Remark 4.** *Simulation results in Figures 3 and 4 show OFUL actions are primarily used in the early stages. This indicates: (1) Greedy-MR and TS-MR implement OFUL actions only when necessary, maintaining a low OFUL fraction throughout most of the time horizon, substantially reducing computational cost; and (2) limited course-corrected exploration at the beginning efficiently remedies TS and Greedy in problematic instances.*

## 8 CONCLUSION

In this work, we propose a data-driven framework to analyze the frequentist regret of POFUL, a family of algorithms that includes OFUL, LinTS, TS-Freq, and Greedy as special cases. Our approach allows for the computation of a data-driven frequentist regret bound for POFUL during implementation, which subsequently informs the course-correction of the algorithm. Our technique conducts a novel real-time geometric analysis of the $d$-dimensional confidence ellipsoid to fully leverage the historical information and might be of independent interest. As applications, we propose TS-MR and Greedy-MR algorithms that enjoy provable minimax optimal frequentist regret and demonstrate their ability to adaptively switch to OFUL when necessary in hard problems where LinTS and Greedy fail. We hope this work provides a steady step towards bridging the gap between theoretical guarantees and empirical performance of bandit algorithms such as LinTS and Greedy.

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
