# OpenReview forum: "Geometry-Aware Approaches for Balancing Performance and Theoretical Guarantees in Linear Bandits"
_ICLR.cc/2025/Conference — ICLR 2025 Poster_

### Official Review · Reviewer_HMrA · 2024-10-27

**Soundness:** 4
**Presentation:** 4
**Contribution:** 4
**Rating:** 8
**Confidence:** 5

**Summary:**

The paper studies algorithms for stochastic linear bandits. Specifically, it has been established that frequentist algorithms may not match the optimal minimax rate in O(d\sqrt{T}) but in practice, “hard” problems are rare and algorithms behave better than what theory predicts. This paper proposes a data-dependent switching strategy that adaptively switches between different “types” of linear bandit algorithms depending on the problem at stake. The switch is made thanks to a threshold that must be tuned.

Beyond this concrete algorithmic contribution, the paper revisit the analysis of linear bandit algorithms and sheds a new light on the problem with a very clear discussion that would be highly valuable to this literature. I recommend accepting this submission to ICLR.

UPDATE: I have changed my score to 8 to also take into account the few remarks raised by the other reviewers and a couple of minor second thoughts I had. However, I still believe this is a clear paper with an original proposal, which I recommend to accept.

**Strengths:**

* Many significant contributions. Concretely, the paper 1/proposes a new way to analyse bandit algorithms that admits all SOTA methods as special cases, 2/ proposes a novel algorithm with an “oracle” regret bound (Th 2) that depends on a problem-dependent quantity, and 3/ shows how to estimate and exploit this quantity for a fully adaptive algorithm, 4/ tests this algorithm on simulated but important examples and on real datasets.
* Well written. The paper is very clear and I only have minor comment further below.
* Impactful. I believe such analysis may have an impact on RL algorithms in Linear MDPs.

**Weaknesses:**

I am honestly not sure what to point out. I was a bit confused by the discussion of \cX \equiv \cS^{d-1} because the notation of the sphere is not defined and I had missed the discussion  on considering only algorithms for which \tau_t=0. But I am not sure what to recommend to make it clearer.

**Questions:**

I don't have specific questions that would be useful for me to assess this work further.

---

> ### Author Response · Authors · 2024-11-23
>
> Thank you for your support of our work and your valuable feedback. We respond to them separately.
> >  I was a bit confused by the discussion of \cX \equiv \cS^{d-1} because the notation of the sphere is not defined
>
> $\mathcal{S}_{d-1}\\coloneqq\lbrace x \in \mathbb{R}^d:\\|x\\|=1\rbrace$ is the unit hypersphere in $\mathbb{R}^d$. We have added the definition as suggested.
>
> >  I had missed the discussion on considering only algorithms for which $\tau\_t=0$
>
> We have revised the discussion of considering only the case where $\tau_t = 0$: Typically, we can check that $\hat{\\alpha}\_{t}$ is no less than one because when $\\tilde{x}\_{t} = x\_{t}^{\\star}$, we have
> $
> \\alpha\_{t} = \\|x\_{t}^{\\star}\\|\_{V\_{t}^{-1}}/\\|\\tilde{x}\_{t}\\|\_{V\_{t}^{-1}} = 1,
> $
> and $\hat{\\alpha}\_{t}$ is an upper bound for $\\alpha\_{t}$. As a result, it holds that
> $
> \\hat{\\mu}\_{t} = (1+\\hat{\\alpha}\_{t})(1+\\iota\_{t}) + (1-\\hat{\\alpha}\_{t})\\tau\_{t} \\leq (1+\\hat{\\alpha}\_{t})(1+\\iota\_{t}),
> $
> i.e., setting $\\tau\_{t} = 0$ yields a valid upper bound for $\hat{\\mu}\_{t}$. Given this observation, we will focus on scenarios where $\\tau\_{t} = 0$ for all $t \\in [T]$. We will add more explanations regarding this in the final version of the proof.

---

> > ### Comment · Reviewer_HMrA · 2024-11-25
> > **Thanks, just a couple of minor comments**
> >
> > Dear authors,
> >
> > Thank you for your answers and updates of the draft.
> >
> > I read all reviews and rebuttals and I want to re-state my strong support for this work. However, I believe my rating may have been a slight over-statement so I'll re-calibrate it a bit to reflect better my general opinion.
> >
> > I think the good performance of Greedy remains quite unknown in the community, especially amongst theorists who don't try it on  "real data". I believe that geometry-adaptive algorithms were missing. I don't know if the right design is to rely on switches such as the idea introduced here, meaning using a threshold that allows to fall back on LinUCB. But this is a good first step and I believe the paper brings good insights.
> >
> > Re-reading the experiments section, I find that the examples 1,2,3 in Figure 3 are a bit too vaguely described in the main text. Perhaps simply by explaining what distinguish them and why example 3 is harder than 2 in some sense. I know them so it did not strike me at first sight, but I think they would deserve to be introduced more clearly, and referred to in the legend of the plots by a name more explicit that "example 1,2,3".

---

> > > ### Author Response · Authors · 2024-11-30
> > >
> > > Thank you again for reviewing our paper and for your valuable support of our work! We will continue revising the paper based on your suggestions and further improve it as we gather additional feedback.

---

### Official Review · Reviewer_fHh6 · 2024-10-31

**Soundness:** 3
**Presentation:** 3
**Contribution:** 3
**Rating:** 8
**Confidence:** 4

**Summary:**

This paper considers the classic stochastic linear bandit problem. Motivated by that the LinTS-freq and Greedy algorithm show great empirical performances but does have minimax optimal regret guarantee, this paper wants to bridge the gap between theoretical guarantees and empirical performance. The paper first proposes a general POFUL algorithm framework which can include LinTS-freq, OFUL, Greedy as special cases and provide a regret bound. Inspired by the analysis, the paper then proposes a Course-Correction version for LinTS-freq and Greedy (TS-MR and Greedy-MR). By adaptively balance the action selection of the original algorithm and OFUL, the paper shows that the proposed algorithm achieves the minimax optimal regret while preserving the original empirical advantage. The better performances of the proposed algorithms are verified through both synthetic and real-world data sets.

**Strengths:**

1.	The paper proposes a general algorithmic framework POFUL that includes classic OFUL, TS, and Greedy as special cases, and provides frequentist regret analysis for this general framework.
2.	The paper introduces the concept of regret proxy, which connects the variance of the optimal arm/selected arms with the regret. Inspired by this, the paper proposes Course-Correction for standard LinTS and Greedy algorithm. The variants combine the empirical advantage of the original algorithms and the theoretical advantage of achieving minimax optimality of OFUL algorithm.
3.	Extensive experiments on both synthetic and real-world datasets are conducted to compare the empirical performances of the classic OFUL/TS/Greedy and the proposed TS-MS and Greedy-MS, which demonstrates consistent advantage over baselines.

**Weaknesses:**

1.	In Line 102, the paper says that “derive a general, instance-dependent frequentist regret bound”. Usually ‘instance-dependent’ in bandit literature means that the regret depends on the sub-optimality gap. Here there may be some confusions.
2.	Generally speaking, the proposed TS-MR and Greedy-MR can be regarded as a combination of traditional TS/Greedy with OFUL. The algorithm adaptively determines which action to follow based on the value of $\hat{\mu}$. Such combination preserves the minimax optimality. However, since the algorithm still needs to compute the OFUL solution, the computational advantage may be limited. Can authors discuss in the experiments what percentage of rounds need to be replaced with OFUL?
3.	Some writing suggestions:
a)	In Section 4.1, the definition of $\tilde{x}_t$ and $x_t^*$ should be moved before the Proposition 4.
b)	In Appendix C, the $\leq$ in Line 872 should be $\geq$, the $\in$ in Line 882 should be $\notin$.

**Questions:**

Please see the last part.

---

> ### Author Response · Authors · 2024-11-23
>
> ### **Weaknesses:**
> > In Line 102, the paper says that “derive a general, instance-dependent frequentist regret bound”. Usually ‘instance-dependent’ in bandit literature means that the regret depends on the sub-optimality gap. Here there may be some confusion.
>
> Thank you for your valuable feedback. We have used “a data-driven frequentist regret bound” instead in the revision to avoid the confusion.
>
> > Since the algorithm still needs to compute the OFUL solution, the computational advantage may be limited. Can authors discuss in the experiments what percentage of rounds need to be replaced with OFUL?
>
> Thank you for your valuable feedback. We have conducted additional simulations (in Appendices I and J of the revised manuscript) to report the percentage of rounds where the OFUL actions are used. The results show that OFUL actions are only frequently used at certain beginning stage of the time horizon, and tends to remain at a low level after that. This indicates that limited amount of course-corrected exploration at the beginning efficiently fixed TS and Greedy in the problematic instances. Hence, although the computational cost depends on the hardness of the problem but is typically low.
>
> > Some writing suggestions: a) In Section 4.1, the definition of $\\tilde{x}_t$ and $x_t^*$ should be moved before the Proposition 4. b) In Appendix C, the $\leq$ in Line 872 should be $\geq$, the $\in$ in Line 882 should be $\notin$.
>
> Thank you for your suggestions. We have implemented these corrections in the revision.

---

> > ### Comment · Reviewer_fHh6 · 2024-11-29
> >
> > Thanks for your response. I have no additional concerns and have increased the score.

---

> > > ### Author Response · Authors · 2024-11-30
> > >
> > > We sincerely thank you for reviewing our paper and increasing the score! Your feedback has guided us to refine and strengthen our work.

---

### Official Review · Reviewer_NV26 · 2024-11-03

**Soundness:** 3
**Presentation:** 4
**Contribution:** 3
**Rating:** 8
**Confidence:** 2

**Summary:**

This paper addresses a significant challenge in stochastic linear bandits, where empirically effective algorithms like Thompson Sampling and Greedy algorithms demonstrate unsatisfying theoretical regret bounds. The author closes this gap by proposing a data-driven approach where problematic instances of LinTS and Greedy are identified, and "course-correction" is applied to those instances. The regret incurred by the new algorithm matches the minimax optimal regret of order $O(d\sqrt{T})$.

**Strengths:**

1. The paper is well written. It follows a logical structure, progressing smoothly from one section to another.
2. Diagrams, such as those illustrating the POFUL algorithm and confidence ellipsoid geometry, help clarify complex ideas, making abstract concepts more tangible. Examples that walk through different algorithms further enhance understanding.
3. The question considered is important. The proposed framework, POFUL, for linear bandit algorithms is novel, supporting OFUL, LinTS, TS-Freq, and Greedy as special cases.
4. The proposed course-correction idea is novel. It adapts LinTS and Greedy algorithms, making them maintain the desirable empirical performance while achieving minimax-optimal frequentist regret.

**Weaknesses:**

1. The method involves setting hyper-parameter $\mu$ (also inflation and optimism parameters) for the course-corrected algorithm. How it is chosen in the experiments is not discussed. More insights into how these parameters influence the outcome or suggestions for selecting optimal values would strengthen the practical utility of the method.
2. Empirically, TS-MR appears to perform similar to the better one of LinTS and OFUL in most cases when the balancing parameter is carefully chosen. While this result aligns with intuition, the paper lacks a discussion of the specific situations in which TS-MR could significantly outperform both LinTS and OFUL.
3. The proposed method involves calculating $\hat{\mu}_t$ each round, which can be can be computationally demanding. This could limit the scalability of the proposed method, especially in applications where the dimension $d$ or the number of actions $T$ is large. It may benefit from a discussion of strategies to mitigate computational costs in such scenarios.

**Questions:**

1. Line 310: "On the right-hand side of equation 4...one cannot apply Proposition 2..." Could you explain why? Proposition 2 applies to an "arbitrary" sequence, so it seems general enough to be applied here.
2. Line 266: why $\tilde{x}_t=x^*_t$ leads to $\hat{\alpha}_t$ no less than 1? If $\hat{\alpha}_t$ is no less than 1, shouldn't the inequality in line 367 go the other way?

---

> ### Author Response · Authors · 2024-11-23
>
> ### **Weaknesses:**
> > The method involves setting hyper-parameters (also inflation and optimism parameters) for the course-corrected algorithm. How it is chosen in the experiments is not discussed. More insights into how these parameters influence the outcome or suggestions for selecting optimal values would strengthen the practical utility of the method.
>
> Thank you for your valuable feedback. In the revised manuscript (Appendix I and J), we have conducted additional simulations to show the influence of $\mu$ and have provided suggestions on tuning it. Furthermore, we should clarify that the inflation and optimism parameters are determined by the choice of algorithm (OFUL, TS, or Greedy). Specifically, for OFUL, we have $\iota_t=0, \tau_t=1$; for Greedy $\iota_t=0, \tau_t=0$; and for LinTS, $\iota_t=\widetilde{\mathcal{O}}(1)$ (which depends on system parameters but remains fixed) and $\tau_t=0$.
>
> > Empirically, TS-MR appears to perform similar to the better one of LinTS and OFUL in most cases when the balancing parameter is carefully chosen. While this result aligns with intuition, the paper lacks a discussion of the specific situations in which TS-MR could significantly outperform both LinTS and OFUL.
>
> Thank you for your valuable feedback. We would like to clarify that the primary goal of the proposed algorithm is not to outperform the baseline algorithms. Rather, it aims to employ only necessary exploration to maintain optimal worst-case performance guarantees while avoiding excessive conservative actions that degrade empirical performance and increase computational costs. Nevertheless, the simulation results show that the course-corrected algorithm can be better than both baseline algorithms (see, e.g. Figure 3(b) and 4(b) in Section 7 where Greedy-MR outperforms Greedy and LinTS).
>
> > The proposed method involves calculating $\hat{\mu}_t$ each round, which can be can be computationally demanding. This could limit the scalability of the proposed method, especially in applications where the dimension $d$ or the number of actions $T$ is large. It may benefit from a discussion of strategies to mitigate computational costs in such scenarios.
>
> Thank you for your valuable feedback. The most computationally demanding step in calculating $\\hat{\mu}\_t$ is the SVD decomposition of the sample covariance matrix $V_t$. However, since $V_t=\lambda_{\text{reg}} I+\sum_{s=1}^t x_s x_s^{\top}$ is updated by a rank-one matrix at each timestep, its SVD decomposition can be updated efficiently (see, e.g., [1]). Consequently, the total complexity of calculating $\hat{\mu}_t$ is equivalent to that of Greedy and TS ($\tilde{O}(Td^2)$) and does not become a computational bottleneck. We will include this discussion in the final version of the paper.
>
> [1] Gandhi, Ratnik, and Amoli Rajgor. "Updating singular value decomposition for rank one matrix perturbation." arXiv preprint arXiv:1707.08369 (2017).
>
> ---
> ### **Questions:**
> > Line 310: "On the right-hand side of equation 4...one cannot apply Proposition 2..." Could you explain why? Proposition 2 applies to an "arbitrary" sequence, so it seems general enough to be applied here.
>
> Thank you for your comments. Note that when implementing the algorithms, the sample covariance matrix at time $t$ takes the form $V_t = \lambda\_{\\text{reg}} I + \sum\_{s=1}^{t} \\tilde{x}\_{s} \\tilde{x}\_{s}^{\\top}$
> ， where $\\{\\tilde{x}\_{s}\\}\_{s \\in [t]}$ are the action chosen at time $s$. Although $\\{\\tilde{x}\_s\\}\_{s\\in[t]}$ can be arbitrary, Proposition 2 holds only when the action sequence and the sample covariance matrix are consistent with each other. Specifically, Proposition 2 can only be applied to $\\{\\tilde{x}\_s\\}\_{s\in[t]}$, yielding the inequality $\\sum\_{s=1}^t \\left\\|\\tilde{x}\_s\\right\\|\_{V\_s^{-1}}^2 \\leq 2 d \\log \\left(1+\\frac{t}{\\lambda\_{\\text{reg}}}\\right).$
>  We will revise the sentence to improve clarity.
>
> > Line 266: why $\tilde{x}_t=x_t^*$ leads to $\alpha_t$ no less than 1? If $\alpha_t$ is no less than 1, shouldn't the inequality in line 367 go the other way?
>
> Thank you for your comments. Since $\\hat{\\alpha}\_t$ serves as an upper bound for $\\alpha\_t$ for all potential $\\tilde{x}\_t$, and given that when $\\tilde{x}\_t = x\_t^{\\star}$, we have $
> \\alpha\_{t} = \\|x\_{t}^{\\star}\\|\_{V\_{t}^{-1}}/\\|\\tilde{x}\_{t}\\|\_{V\_{t}^{-1}} = 1$, it follows that $\\hat{\\alpha}\_t \\geq 1$. We have corrected the inequality typo in line 367 and revised the accompanying discussion.

---

> > ### Comment · Reviewer_NV26 · 2024-11-27
> > **Thank you for feedback**
> >
> > Since most of my concerns have been addressed, I have raised my score.

---

> > > ### Author Response · Authors · 2024-11-30
> > >
> > > Thank you again for your efforts in reviewing our paper and for raising the score! Your valuable feedback has been instrumental in improving the quality of our work.

---

### Official Review · Reviewer_jV72 · 2024-11-03

**Soundness:** 3
**Presentation:** 3
**Contribution:** 2
**Rating:** 6
**Confidence:** 4

**Summary:**

This paper studies the $d$-dimensional stochastic linear bandit setting. It builds on several previous works, mostly on
- the analysis from Abbasi-Yadkori et al. (2011) where they proposed an algorithm "Optimism in the Face of Uncertainty  Linear Bandit Algorithm" (OFUL) for which they derived a frequentist regret bound in $\tilde{O}(d\sqrt{T})$;
- the work from Agrawal \& Goyal (2013) and Abeille et al. (2017) that studied the frequentist variant of LinTS, referred to as TS-Freq, and showed that its frequenist regret is upper bounded in $\tilde{O}(d^{3/2}\sqrt{T})$. The extra factor $d^{1/2}$ has latter confirmed to be necessary by Hamidi \& Bayati (2020);
- the works from Russo \& Van Roy (2014) and Dong \& Van Roy (2018), who studied the Bayesian regret of the Thompson Sampling algorithm, referred to as LinTS and proved that it is upper bounded in $\tilde{O}(d\sqrt{T})$.

Based on the last points regarding LinTS, the authors conclude that the algorithm performs well in most problem instances but suffers in unfavorable, unlikely settings. Their idea is then to propose a modified "course-corrected" version of LinTS and Greedy algorithms, that can switch to selecting OFUL actions under a criteria based on a geometric analysis the $d$-dimensional confidence ellipsoid. The resulting algorithms are called respectively "Linear
Thompson Sampling with Maximum Regret (Proxy)" (TS-MR) and Greedy-MR. Both algorithms retain the theoretical guarantees of OFUL algorithm while presenting some computational and performance advantages from the LinTS and Greedy method.

**Strengths:**

The main strength of the paper is to propose a modification of the LinTS and Greedy algorithms, two well studied algorithms for solving $d$-dimensional stochastic linear bandit problems. The modification is based on the work from Abbasi-Yadkori et al. (2011) and their algorithm "Optimism in the Face of Uncertainty  Linear Bandit Algorithm" (OFUL) which enjoys frequentist regret bound in $\tilde{O}(d\sqrt{T})$. The authors resulting algorithms "Linear
Thompson Sampling with Maximum Regret (Proxy)" (TS-MR) and Greedy-MR, can both switch from their respective methods to OFUL algorithm under some criteria. The authors derive frequentist regret bound in $\tilde{O}(d\sqrt{T})$ for both algorithms and illustrate their performance under some numerical experiments.

**Weaknesses:**

Although the main ideas of the paper are interesting, there are a few points that could be improved, such as listed below.

- The main weakness concerns the reproducibility of the experiments. The authors provide few explanations regarding the experiment setup in the paper and no information regarding the algorithm's parameters and implementation. They did not provide their code, and the details they give in the Appendix are not sufficient to reproduce the same experiments. Therefore, it was impossible to assess the soundness of the simulation results.
- A second weakness concerns the experimental comparison. The authors should have included experiments assessing the influence of the algorithm parameter $\mu$ on the performance. They authors mention that they "aim to compare TS-MR, Greedy-MR, and key baseline algorithms, via simulation" but the key baseline algorithm they compared their method with are the very same algorithm their method is based upon. It would have been for the authors to include a comparison with a different algorithm such as UCB.
- Another point of improvement concerns the figures in the paper. First, the font size in all figures in the paper is too small to read on a printed version of the paper. Second, some figures lack explanations such as Figure 3 that seems to show confidence interval for the regret of Greedy algorithm and LinTS algorithm without any explanation.

**Questions:**

Here is a list of suggestions for the authors.
- A first suggestion is to include a proof of Corollary 1, if possible, in the main part of the paper.
- A second suggestion is to include the pseudocode of the proposed methods TS-MR and Greedy-MR in the paper's main text.
- Another suggestion concerns the experiments. The authors should provide enough material and explanations so that it is possible to replicate and verify the results of the simulations. We suggest the authors to share the code they used to generate the simulations, as well as to include more details about the experiments in the Appendix:
  - the pseudocode and parameters used for the TS-Freq algorithm
  - the pseudocode and parameters used for the OFUL algorithm
  - the pseudocode and parameters used for the Greedy algorithm
  - the values of the parameters ($\delta$, $\iota_t$, $\lambda$) for the TS-MR algorithm as well as explanations regarding the distribution $\mathcal{D}^{SA}(\delta')$.
    The authors should also include references and explanations regarding "the standard approach in the literature that converts classification tasks to contextual bandit problems". Note that there is a typo on line 500 as there is no Example 2 in Section 6.
    Ideally, the authors should also perform experiences to illustrate the tuning of the parameter $\mu$ and its influence on the performance of the algorithm and include comparisons to algorithms that are not part of the proposed method, such as UCB.
- The authors should also improve the readability of the Figures in the main body of the paper as the font size is too small. They should also provide explanations in Figures 3 (b) and (C) regarding what seems to be confidence intervals.
- There are some notations throughout the paper that are either not defined or used before they were defined:
  - the notations $\lambda_1,\ldots,\lambda_d$ and $\lambda_2^{-\frac{1}{2}}(V_t)$ used in Theorem 2 but are not defined. There is also some confusion with both the notation $\lambda$ being used as a regularization parameter. We note that both the notation $\lambda_d$ and $\lambda_d(V_t)$ are present in the paper. It is not clear if it refers to the same quantity.
  - there seems to be a typo on line 423 in Theorem 2, as the indexes $k$ and $k+1$ are used but do not refer to anything. The authors should bring clarification.
  - the function $x^\star_t(\theta)$ is used on line 351 but only defined on line 383.
  - the notation $\equiv$ is used but not defined.
  - the notation $\mathcal{S}_{d-1}$ is used but not defined. It is unclear whether it refers to the $d$-dimensional unit sphere or a sphere of radius $S$ centered on the origin.
  - the notation $\beta^{RLS}_t(\delta)$ hides the dependency on the parameter $\lambda$.
  - it is not explained how $\hat{\mu}_t$ relates to $\hat{\alpha}_t$ on line 364.
- Other suggestions include:
   - avoiding abbreviations such as "we'll" on line 368 and prefer the full expression
   - citing reference on line 370 to explain the "large regret proxies and computational inefficiencies seen in OFUL when determining the action".
   - it is unclear from Example 2 how to set the variables $\iota^{TS}$ such that POFUL matches LinTS.
   - Assumption 3 could be renamed to "Subgaussian reward noise."
   - rephrase line 390, "As is illustrated on Figure 2." as this phrase lacks a subject and is therefore not a proper sentence.
   - correct the typo on line 1048: "We ser $\mu$".

---

> ### Author Response · Authors · 2024-11-23
>
> ### **Weaknesses:**
> > The main weakness concerns the reproducibility of the experiments. The authors provide few explanations regarding the experiment setup in the paper and no information regarding the algorithm's parameters and implementation. They did not provide their code, and the details they give in the Appendix are not sufficient to reproduce the same experiments. Therefore, it was impossible to assess the soundness of the simulation results.
>
> Thank you for your comments. We have attached the code to the supplementary material to reproduce the simulation results.
>
> > A second weakness concerns the experimental comparison. The authors should have included experiments assessing the influence of the algorithm parameter on the performance. ... It would have been for the authors to include a comparison with a different algorithm such as UCB.
>
>
> - We appreciate the reviewer's comment. We have conducted simulations in the revised manuscript ( Appendices I and J ) to illustrate the influence of $\mu$ and have provided suggestions on tuning it.
> - We appreciate the reviewer's comment, which provides us with the opportunity to further clarify the relationship between UCB and OFUL in the context of linear bandit problems.  At a high level, OFUL implements the UCB principle by utilizing the linear structure of the problem and can be regarded as a UCB-type algorithm for linear bandits. Specifically, UCB chooses the action with the highest upper confidence bound, while OFUL solves: $\max\_{x \in \mathcal{X}\_t} \\left\\langle x, \\widehat{\\theta}\_t \\right\\rangle + \\|x\\|\_{V\_t^{-1} }\\beta\_{t,\\delta^{\\prime},\\lambda_{\\text{reg}}}^{RLS}$
> , and here $\\|x\\|\_{V\_t^{-1} }\\beta\_{t,\\delta^{\\prime},\\lambda_{\\text{reg}}}^{RLS}$ represents the confidence radius of the reward of $x$. We will add further explanation regarding this to improve clarity in the final version of the paper.
>
> >Another point of improvement concerns the figures in the paper. First, the font size in all figures in the paper is too small to read on a printed version of the paper. Second, some figures lack explanations such as Figure 3 which seems to show a confidence interval for the regret of Greedy algorithm and LinTS algorithm without any explanation.
>
> We appreciate the reviewer’s feedback. We have adjusted the font size of the figures to improve the readability. The confidence intervals in Figure 3 show the $\pm$ 2*SE of the mean regret. We will add more explanation in the final version of the paper.
>
> ---
> ### **Questions:**
> > A first suggestion is to include a proof of Corollary 1, if possible, in the main part of the paper.
>
> We appreciate your feedback and have accordingly incorporated the proof.
>
> > A second suggestion is to include the pseudocode of the proposed methods TS-MR and Greedy-MR in the paper's main text.
>
> We appreciate your feedback feedback and will include them in the final version of the paper.
>
> > Another suggestion concerns the experiments. The authors should provide enough material and explanations so that it is possible to replicate and verify the results of the simulations. We suggest the authors share the code they used to generate the simulations, as well as to include more details about the experiments in the Appendix: ...
>
> Thank you for your valuable feedback.
> - We have attached the code to the supplemental material to fully reproduce our simulation results and will add the pseudocode to the final version of the paper.
> - We have included the references regarding converting classification tasks into bandit problems.
> - For all simulations,  we set $\lambda_{\text{reg}} = 1$, $\delta = 0.001$ and sample $\\eta_t \\sim \\mathcal{D}^{SA}(\\delta) = \\mathcal{N}(0, \\frac{1}{\\sqrt{2d \\log(2d/\\delta)}}I_d) $ such that $\\mathbb{P}_{\\eta \\sim \\mathcal{D}^{S A}(\\delta)}[\|\\eta\| \leq 1] \\geq 1-\\delta$. For TS-MR, we set $\iota_t = 1$ for all $t \in [T]$.
> - We have conducted simulations in the revised manuscript ( Appendices I and J ) to illustrate the influence of $\mu$ and have provided suggestions on tuning it.
> - We have fixed the typo on line 500.
>
> > The authors should also improve the readability of the Figures in the main body of the paper as the font size is too small. They should also provide explanations in Figures 3 (b) and (c) regarding what seems to be confidence intervals.
>
> We appreciate your feedback and have adjusted the figures accordingly. We will add more explanations in the final version of the proof.
>
>
> > There are some notations throughout the paper that are either not defined or used before they were defined: ...
>
> Thank you for your suggestions. We have fixed all the notation problems mentioned above in the revision.
>
> > Other suggestions include: ...
>
> Thank you for your suggestions. We have modified all the typos mentioned.

---

> > ### Comment · Reviewer_jV72 · 2024-12-01
> >
> > Thank you for your response. As most of my concerns have been addressed, I have increased my score.

---

> > > ### Author Response · Authors · 2024-12-01
> > >
> > > Thank you so much for reviewing our work and increasing the score! Your insightful and constructive feedback has been crucial in enhancing the quality of our work.

---

### Official Review · Reviewer_5WZb · 2024-11-29

**Soundness:** 3
**Presentation:** 3
**Contribution:** 3
**Rating:** 5
**Confidence:** 3

**Summary:**

This paper studied the classical linear bandit problem. It proposed the POFUL algorithm which can be specified to be LinTS, LinUCB and greedy algorithms with different parameters respectively. It analyzed its regret and evaluated its performance with numerical experiments. This paper also considered the continuous action space and the instances with proxy.


======

After rebuttal: I prefer to keep the score.

**Strengths:**

1. This paper provided a more general algorithm for regret minimization in linear bandits and another aspect to understand the efficient algorithms.
1. The numerical experiments are well designed as different types of bandit examples are considered.

**Weaknesses:**

1. The author(s) may consider to place the comparison among Theorem 1 and results from existing papers in the main paper. The comparison is not clear at first glance.
1. Sections 6 and 7 indicate that a preset threshold $\mu$ can clearly reduce the regret and Remark 3 discussed the choice of $\mu$. However, I wonder under which types of real-life scenarios we can know $\mu$ and what is the choice of $\mu$ in simulations.

**Questions:**

Please refer to the *Questions* section.

---

> ### Author Response · Authors · 2024-11-30
>
> We appreciate the reviewer for the constructive feedback. We respond to them separately below.
> > The author(s) may consider to place the comparison among Theorem 1 and results from existing papers in the main paper. The comparison is not clear at first glance.
>
> Thank you for your suggestion. We will enhance the comparison related to Theorem 1 and include it in the main text of the final version of the paper.
>
> > Sections 6 and 7 indicate that a preset threshold $\mu$ can clearly reduce the regret and Remark 3 discusses the choice of $\mu$. However, I wonder under which types of real-life scenarios we can know $\mu$ and what is the choice of $\mu$ in simulations.
>
> Thank you for your comment. For synthetic datasets, we set $\mu = 8$ in Example 1 for both TS-MR and Greedy-MR algorithms. In Examples 2 and 3, we set $\mu = 12$ for both algorithms. For all three real-world datasets, we used $\mu = 10$ for TS-MR and $\mu = 12$ for Greedy-MR. We have attached the code to the supplementary materials for reproducibility, and we will incorporate these simulation details into the main text in the final version of the paper for clarity.
>
> Additionally, we have conducted simulations to address the concern regarding choosing $\mu$. Specifically, in Appendix I of the revised manuscript, we report the fraction of OFUL actions used throughout the entire time horizon for all simulations. In Appendix J of the revised manuscript, we illustrate the impact of $\mu$ on algorithm performance by varying its value and provide a detailed discussion of its tuning.
>
> The new simulation results show that:
>
> 1. OFUL actions are primarily implemented in the early stages of the time horizon, with their fraction remaining low in subsequent stages. This indicates that proper exploration during the initial stage benefits the algorithms' long-term performance.
> 2. The course-corrected algorithm exhibits robust performance across different choices of $\mu$. Our simulations show that setting $\mu\in[8,12]$ performs well across all simulated scenarios.
> 3. In practice, we recommend setting $\mu$ to a moderate value, typically within the range $[8,12]$, as validated by our simulation results. The selection of $\mu$ should aim to ensure effective course-corrected exploration during the initial stage: A suitable $\mu$ value is indicated when the fraction of OFUL actions is high at the beginning and gradually decreases to and maintains a low level. If the fraction of OFUL actions remains consistently high, increasing $\mu$ can help reduce computational costs. Conversely, if very few OFUL actions are executed during the initial stage, decreasing $\mu$ helps with exploration. Finally, since algorithmic performance remains robust across moderate values of $\mu$, the precise selection of $\mu$ is unlikely to be a significant practical concern.
>
> We hope our responses have addressed your concerns, and we would be happy to answer any further questions you may have.

---

> > ### Author Response · Authors · 2024-12-01
> >
> > We would like to kindly remind the reviewer that the deadline for posting comments to the authors is Dec. 2 AoE, while the authors’ deadline to respond is Dec. 3 AoE. If you have any additional concerns or suggestions, we would greatly appreciate the opportunity to address them before the deadline.
> >
> > Thank you so much for your time and effort in reviewing our work.

---

> > > ### Comment · Reviewer_5WZb · 2024-12-02
> > >
> > > Thanks for your response.
> > > 1. At this moment, I am still not sure how Theorem 1 shows the proposed algorithm is better than those existing algorithms and may keep my score.
> > > 1. Regarding $\mu$: the author(s) may consider to include the discussions in the paper.

---

> > > > ### Author Response · Authors · 2024-12-02
> > > >
> > > > Thank you for your reply. We address your points in detail below.
> > > >
> > > > > Comparison between Theorem 1 and results from existing literature.
> > > >
> > > > Regarding the frequentist setting considered in this work, the results in the literature are as follows:
> > > >
> > > >  - The OFUL algorithm proposed by [1] achieves a frequentist regret bound of $\tilde{\mathcal{O}}(d\sqrt{T})$, which is optimal up to logarithmic factors, as shown by [2].
> > > > - For the LinTS algorithm, [3] and [4] demonstrate that by inflating the posterior of LinTS by a factor of $\sqrt{d}$ (resulting in the TS-Freq algorithm discussed in the paper), a frequentist regret bound of $\tilde{\mathcal{O}}(d\sqrt{dT})$ can be derived.
> > > > - However, for the original LinTS algorithm and the Greedy algorithm, no frequentist regret guarantee exists in the literature. Moreover, [5] shows that these algorithms fail in certain bad instances.
> > > >
> > > > We make the following comparison between our Theorem 1 and the existing results.
> > > > - **Generality of Theorem 1**: Our Theorem 1 provides a frequentist regret bound for the POFUL algorithm family, which encompasses OFUL, LinTS, TS-Freq, and Greedy as special cases. Therefore, it serves as a valid bound for all these algorithms.
> > > >
> > > > - **Recovering Standard Regret Bounds**: By substituting the corresponding $\mu_t$ values, standard regret bounds can be derived from our Theorem 1:
> > > >     - For the OFUL algorithm, by design, $\mu_t = 2$ for all $t \in [T]$. Thus, applying Theorem 1 yields a regret bound of $\tilde{\mathcal{O}}(d\sqrt{\sum_{t = 1}^T \mu_t^2 }) = \tilde{\mathcal{O}}(d\sqrt{\sum_{t = 1}^T 4 }) = \tilde{\mathcal{O}}(d\sqrt{T})$, recovering the bound in [1].
> > > >
> > > >     - For the TS-Freq algorithm, where the inflation parameter is $\iota_t = \tilde{\mathcal{O}}(\sqrt{d})$ (due to posterior inflation), and the optimism parameter is $\tau_t = 0$ for all $t \in [T]$, we have $\mu_t = \alpha_t (1 + \iota_t - \tau_t) + 1 + \iota_t + \tau_t = (1 + \alpha_t)(1 + \iota_t) = (1 + \alpha_t)\tilde{\mathcal{O}}(\sqrt{d})$. Applying Theorem 1 yields a regret bound of $\tilde{\mathcal{O}}(d\sqrt{\sum_{t = 1}^T \mu_t^2}) = \tilde{\mathcal{O}}(d\sqrt{d\sum_{t = 1}^T (1 + \alpha_t)^2})$. This matches the bounds in [3] and [4] when $\alpha_t = \tilde{\mathcal{O}}(1)$, which can be validated by simulation results.
> > > >     - For the original LinTS and Greedy algorithms, which fail in certain bad instances as shown in [5], our Theorem 1 remains valid: In such cases, $\mu_t$ can exceed $O(1)$, accurately reflecting the failure.
> > > >
> > > > - **Implications and Algorithm Design**: Theorem 1 further suggests that if $\mu_t = \tilde{\mathcal{O}}(1)$ for all $t \in [T]$, the optimal regret bound of $\tilde{\mathcal{O}}(d\sqrt{T})$ can be achieved. Inspired by this, we propose TS-MR and Greedy-MR, which enforce a threshold $\mu$ for $\mu_t$ in LinTS and Greedy. When the estimated upper bound $\hat{\mu_t}$ exceeds $\mu$, these algorithms switch to OFUL actions, ensuring that $\mu_t = \tilde{\mathcal{O}}(1)$ for all $t \in [T]$. Consequently, TS-MR and Greedy-MR attain the optimal regret bound.
> > > >
> > > > We will enhance and include the comparison for improved clarity in the final version of the paper.
> > > >
> > > > > Regarding $\mu$: the author(s) may consider to include the discussions in the paper.
> > > >
> > > > Thank you for your suggestion and we will include them in the main text of the final version of the paper.
> > > >
> > > > ---
> > > >
> > > >
> > > > [1] Abbasi-Yadkori, Yasin, Dávid Pál, and Csaba Szepesvári. "Improved algorithms for linear stochastic bandits." Advances in neural information processing systems 24 (2011).
> > > >
> > > > [2] Dani, Varsha, Thomas P. Hayes, and Sham M. Kakade. "Stochastic Linear Optimization under Bandit Feedback." COLT. Vol. 2. 2008.
> > > >
> > > > [3] Agrawal, Shipra, and Navin Goyal. "Thompson sampling for contextual bandits with linear payoffs." International conference on machine learning. PMLR, 2013.
> > > >
> > > > [4] Abeille, Marc, and Alessandro Lazaric. "Linear thompson sampling revisited." Artificial Intelligence and Statistics. PMLR, 2017.
> > > >
> > > > [5] Hamidi, Nima, and Mohsen Bayati. "On frequentist regret of linear thompson sampling." arXiv preprint arXiv:2006.06790 (2020).

---

### Author Response · Authors · 2024-11-23

We sincerely thank the reviewers for their valuable and constructive feedback. In response, we have made several significant revisions to our submission:
- First, we have addressed the shared concerns regarding the hyper-parameter $\mu$ — specifically its tunning, its impact on algorithmic outcomes, and the corresponding fraction of OFUL actions implemented—we have conducted additional simulations, which are now included in Appendices I and J of the revised manuscript.
- Second, we have incorporated suggested revisions and have fixed the typos and notation issues. These modifications are highlighted in blue in the revised manuscript.
- Finally, we have included our code in the supplementary materials to ensure full reproducibility of our simulation results.

Below, we address each point of feedback respectively. We trust our responses adequately address the reviewers' concerns. We would also be more than happy to address any additional questions the reviewers may have.

---

### Meta-Review · Area_Chair_CEA4 · 2024-12-20

**Metareview:**

The paper studies algorithms for stochastic linear bandits and proposes a data-dependent switching strategy that adaptively switches between different linear bandit algorithms depending on the problem at stake. Important contributions are the derivation of a problem-dependent quantity that can exploited to derive fully adaptive algorithm which adjust to the problem at stake.
This will in all likelihood turn out to be an impactful paper in the bandit community. Reviews are overall very positive and the paper has a high average. There are a variety of suggestions by the reviewers of how to improve the paper and I urge the authors to implement these.

**Additional Comments On Reviewer Discussion:**

There was a useful discussion and review ratings were adjusted afterwards.

---

### Decision · Program_Chairs · 2025-01-22

Accept (Poster)